# Predicting ruminal degradability and chemical composition of corn silage using near-infrared spectroscopy and multivariate regression

**Pauliane Pucetti**[1]*, **Sebastião de Campos Valadares Filho**[1], **Jussara Valente Roque**[2], **Julia Travassos da Silva**[1], **Kellen Ribeiro de Oliveira**[1], **Flavia Adriane Sales Silva**[1], **Wilson Junior Cardoso**[2], **Fabyano Fonseca e Silva**[1], **Kendall Carl Swanson**[3]

1 Department of Animal Sciences, Universidade Federal de Viçosa, Viçosa, Minas Gerais, Brazil,
2 Department of Chemistry, Universidade Federal de Viçosa, Viçosa, Minas Gerais, Brazil, 3 Department of Animal Sciences, North Dakota State University, Fargo, North Dakota, United States of America

* pauliane.pucett@ufv.br

**Data Availability Statement:** All relevant data are within the paper and its Supporting information files.

## Abstract

The aim of this study was to develop and validate regression models to predict the chemical composition and ruminal degradation parameters of corn silage by near-infrared spectroscopy (NIR). Ninety-four samples were used to develop and validate the models to predict corn silage composition. A subset of 23 samples was used to develop and validate models to predict ruminal degradation parameters of corn silage. Wet chemistry methods were used to determine the composition values and ruminal degradation parameters of the corn silage samples. The dried and ground samples had their NIR spectra scanned using a poliS-PECNIR 900–1700 model NIR sprectrophotometer (ITPhotonics S.r.I, Breganze, IT.). The models were developed using regression by partial least squares (PLS), and the ordered predictor selection (OPS) method was used. In general, the regression models obtained to predict the corn silage composition (P>0.05), except the model for organic matter (OM), adequately estimated the studied properties. It was not possible to develop prediction models for the potentially degradable fraction in the rumen of OM and crude protein and the degradation rate of OM. The regression models that could be obtained to predict the ruminal degradation parameters showed correlation coefficient of calibration between 0.530 and 0.985. The regression models developed to predict CS composition accurately estimated the CS composition, except the model for OM. The NIR has potential to be used by nutritionists as a rapid prediction tool for ruminal degradation parameters in the field.

## Introduction

Corn silage (CS) is an important source of nutrients, particularly energy and digestible fiber, in modern dairy cattle operations worldwide and beef cattle operations in Europe, Latin America, and North America [1–5]. Many factors contribute to the high use of CS by producers, including lower harvesting costs, minimized risks of production, elevated yield per area, and flexibility to harvest corn for forage or grain [6]. However, CS quality is influenced

**Funding:** This work was supported by the National Council of Scientific and Technological Development (CNPq - Grant Number - 465377/2014-9), National Institute of Science and Technology in Animal Science (INCT - Ciência Animal - Grant number - 465377/2014-9), and Coordination of Improvement of Personal Higher Education (CAPES, PROEX: 32002017011P9). The funding agency had no role in the study design, data collection, and analyses, decision to publish, or preparation of the manuscript. The funders had no role in study design, data collection and analysis, decision to publish, or preparation of the manuscript.

**Competing interests:** The authors have declared that no competing interests exist.

by several factors, such as production technique, type of corn hybrid (e.g., flint corn), and climate factors. Thus, the use of tabular values of CS chemical composition found in the literature may not be appropriate to formulate ruminant diets. However, conventional laboratory analyses to measure dietary ingredient chemical composition are laborious and time-consuming.

In addition to feed chemical composition values, nutritional requirement systems for ruminants use ruminal degradability of nutrients to formulate diets with greater precision [7–9]. For example, measuring the ruminal degradability of dietary crude protein allows for predicting the supply of metabolizable protein, which is the total amount of true protein available for intestinal absorption. Additionally, there are models to predict milk yield that consider the effect of NDF degradability [7].

The *in situ* nylon bag technique is a widely adopted procedure to characterize the dynamics of degradation of feedstuffs and nutrients in the rumen. However, animal ethics committees have recommended stricter animal use protocols and the reduction or, if possible, the replacement of animals with alternative laboratory methods in research settings [10,11]. In addition, the costs of animals, surgical supplies, feeding, and labor are important limitations to these degradability measurements.

In this context, near-infrared (NIR) spectroscopy is an alternative approach to access feed composition and ruminal degradability that does not require the use of animals or chemical analyses. Furthermore, it requires little or no previous sample preparation. Near-infrared spectroscopy has been routinely used for the nutritional analysis of silage and other livestock feeds in the dairy and beef industries and the procedure is rapid and inexpensive as compared to *in situ* or *in vivo* measurements [12]. Less information is available accessing the rate and extent of ruminal degradability of nutrients using NIR approaches.

Some published studies demonstrate the potential of NIR spectroscopy to predict feed analysis [12–14]. However, these studies utilized the full-spectrum regression methodology, which do not incorporate variable selection methods that can prevent the use of irrelevant or redundant variables as well as variables that represent noise, commonly observed in large databases such as those used in NIR spectra studies[15–17].

Thus, we hypothesized that regression models can accurately predict both the composition and ruminal degradability parameters of CS produced in Brazil. Hence, the objectives of this study were to develop and evaluate regression models using the ordered predictor selection (OPS) approach to predict the chemical composition and ruminal degradation parameters of CS by NIR spectra.

## Material and methods

All procedures were previously approved by the Animal Ethics and Welfare Committee of the Universidade Federal de Viçosa (#042/2019).

### Sample collection and preparation

Overall, 94 CS samples were collected in eight states of Brazil (Bahia, Goiás, Minas Gerais, Mato Grosso, Pernambuco, Paraná, Rio Grande do Sul, and São Paulo) to provide sufficient variation of the CS feeding value to develop and evaluate the models. The samples were sent to the Ruminant Nutrition Laboratory of the Animal Science Department at the Universidade Federal de Viçosa (UFV), Viçosa, Minas Gerais, Brazil for processing and performing typical laboratory analyses. Samples were dried in a forced air oven (55˚C) for 72 hours and ground to pass a 2-mm and 1-mm screen (Tecnal, Piracicaba, São Paulo, Brazil).

## In situ degradation procedures

*In situ* degradability measurements were conducted at the Ruminant Nutrition Laboratory of the Animal Science Department at the Universidade Federal de Viçosa, Viçosa, Minas Gerais, Brazil.

The 23 samples were divided into 3 groups and ruminally incubated in three Nellore bulls, with an average BW of 450 ± 13 kg and 24 mo of age in a $3 \times 3$ Latin square design. Of the three corn silage groups, two contained eight corn silage samples, and one group contained seven corn silage samples. Within each period, each corn silage group was incubated in the rumen of a different bull. As reported by Machado et al. [18], the objective of the Latin square was to assist with the organization of the information that was collected in the field while allowing for measurements of degradation of different feeds without confounding the effect of the animal as well as to control sources of variation and avoid bias without estimating the variability.

The bulls were fed for ad libitum intake a 70:30 (DM basis) corn silage:concentrate (corn, soybean meal, wheat bran, urea, ammonium sulfate, salt, bicarbonate, and mineral) diet. The diet contained 12% crude protein (DM basis). The forage-to-concentrate ratio used was based on providing a balanced diet with diverse feed composition with enough fiber and energy to not compromise microbial growth. The animals were adapted to the experimental diet and conditions for 30 d before the *in situ* incubations.

Individually identified nylon bags (Sefar Nitex; Sefar AG, Thal, Switzerland; porosity of 50 μm-CA and 8 by 15 cm) were used containing 6.0 g of each sample that was previously ground using a 2-mm screen. Incubation times were 0, 3, 6, 12, 24, 48, 72 and 96 h. The number of bags varied as a function of the incubation time to guarantee enough residual sample after incubation (i.e. more bags per sample were incubated for the longer incubation times relative to the shorter incubation times). In situ bags containing samples were attached to a steel chain ($90 \times 2$ cm) with a weight at the end, thus allowing for complete immersion within the ruminal fluid, below the fiber mat. Bags were placed into the rumen in reverse order of incubation hours such that all bags were removed at the same time for rinsing. After the incubation period, the bags were rinsed in running water followed by washing with cold tap water by hand by the same person. The endpoint for rinsing was when the rinse water was clear after flow through the bags [19]. The 0-h bags were not incubated in the rumen but were rinsed using the same procedures for rinsing as described above. Samples were oven-dried at 55˚C for 72 h. Bags were placed in an oven at 105˚C for 2 h and weighed. The residue was removed from the nylon bags, ground in a knife mill (Tecnal, Piracicaba, São Paulo, Brazil) with a 1-mm sieve, and placed in a labeled plastic bag.

## Analytical methods for reference data

Samples were analyzed according to standard analytical methods of the Brazilian National Institute of Science and Technology in Animal Science [20] for DM and DM at 105˚C (method INCT—CA G-003/1), ash (complete combustion in a muffle furnace at 600˚C; method INCT-CA M-001/1), CP (Kjeldahl procedure; method INCT-CA N-001/1), EE (Randall extraction; method INCT-CA G-005/1), NDF (using a heat-stable α-amylase, omitting sodium sulfite; method INCT-CA F-001/1), ADF (method INCT–CA F-003/1), and lignin (method INCT-CA F-005/1). The OM concentrations were estimated by the difference between the DM and ash concentrations. The NDF and ADF were corrected for residual N compounds (NDIP = method INCT—CA N-004/1; and ADIP = method INCT—CA N-005/1) and ash (method INCT—CA M002/1 and method INCT—CA M003/1). The iNDF concentrations was evaluated as the residual NDF remaining after 288 h of ruminal *in situ* incubation according to

Casali et al. [21]. Starch concentrations were analyzed following the method described by Zinn [22] and modified by Silva et al. [23]. Sample NFC concentrations were analyzed as suggested by Detmann and Valadares Filho [24] as NFC = OM—CP—apNDF—EE, where apNDF is the NDF corrected for ash and residual N compounds, as described above. The residues from in situ incubations were analyzed for DM, OM, CP, and NDF by the methods described above.

### *In situ* degradation parameters

*In situ* degradation parameters for DM, OM, and CP were estimated by using the first-order asymptotic model described by Ørskov and McDonald [25]:

$$Y(t) = a + b \times \left( e^{-kd \times t} \right)$$

where Yt = degraded fraction of DM, OM or CP in time 't', (%); a = readily soluble fraction, (%); b = potentially degradable fraction in the rumen, (%); kd = rate constant for degradation of b, per h; and t = time, h.

The NDF degradation parameters were estimated by using the model proposed by Van Milgen et al. [26]

$$RNDF(t) = b \times (1 + \lambda \times t) \times e^{-\lambda * t} ) + I$$

where RNDFt = non-degraded NDF residue at time "t" (%); b = potentially degradable fraction in the rumen (%); $\lambda$ = joint fractional rate of latency and degradation (h$^{-1}$); t = time independent variable (h); I = undegradable fraction (%).

The calculation for the equation used for DM and OM is for the degraded fraction and the calculation for the equation used for NDF is for the non-degraded residue. Therefore, NDF degradation was calculated by difference: Degradation (%) = 100 –residue.

The NDF degradation rate was calculated based on $\lambda$, using the properties of the $\Gamma(2)$ distribution [27]:

$$kd = 0,59635 \times \lambda$$

where kd = rate constant for degradation of b (% per h); $\lambda$ = joint fractional rate of latency and degradation (h$^{-1}$). The parameters "a", "b", kd, $\lambda$, and "I" of the *in situ* incubation models were estimated using the PROC NLIN procedure (version 9.4, SAS Institute Inc., Cary, NC, USA), and assuming the Marquardt algorithm for convergence.

### Acquisition of spectral data

Ground CS samples were homogenized, placed in Petri dishes (60 mm diameter), and scanned at 2-nm intervals from 902 to 1,680 nm, using a poliSPECNIR 900–1700 model NIR sprectrophotometer (ITPhotonics S.r.l, Breganze, IT.). Spectra were recorded three times, averaged, and stored as the logarithm of the reciprocal of reflectance (1/R). The PoliDATA software (ITPhotonics S.r.l, Breganze, IT) was used to acquire spectral data.

### Regression models

The NIR spectra data were exported as an.*xls* file using the SensorLogic GmbH software (Software + Sensor Systeme, Norderstedt, Germany) and imported by the Matlab 2019b software (Math Works, Natick, USA). A data matrix with the NIR spectra was built and named **X** matrix for each evaluated property, which represented the independent variables. The rows of the **X** matrix corresponded to the samples, and the columns corresponded to the variables (wavelength).

Twenty-seven properties were determined and identified as y vectors (dependent variables). The properties evaluated to estimate the CS composition were DM, DM at 105˚C, OM, CP, NDIP, ADIP, EE, NDF, apNDF, iNDF, ADF, apADF, lignin, NFC, and starch. The *in situ* degradation parameters evaluated to estimate ruminal degradation of CS were the "a" of DM, OM, and CP, "b" and "kd" of DM, OM, CP, and NDF, and "I" of NDF. The **y** vector had a length equal to the number of rows in the **X** matrix. For each property, a dataset was prepared.

Post-outlier removal, the dataset was then divided into calibration and prediction sets using the Kennard-Stone algorithm [28]. The Kennard-Stone algorithm is advantageous as it aids in the creation of representative subsets for model development, contributing to enhanced predictive accuracy. Additionally, the Kennard-Stone algorithm is known for its computational efficiency, making it a practical choice for large datasets. The data set for ruminal degradation of CS was not split, because of the limited number of samples evaluated, and the model was evaluated by cross-validation. The **y** variable was mean-centered for all properties evaluated and different pre-treatments were studied for each **X** matrix. The pre-treatments tested were mean centering, autoscale, smoothing, first and second derivative, multiplicative scatter correction (MSC), normalize, baseline, and standard normal variate (SNV), and detrend. Combinations of two, three, and four pre-treatments were also evaluated. The best pre-treatment for each property was chosen based on the lowest root mean square error of cross-validation ($RMSE_{CV}$).

Partial least squares regression (PLS) was used to develop the models using variables selected through three approaches of the OPS method: automatic OPS (autoOPS), feedback OPS (feedOPS), and OPS intervals (iOPS) as described by Roque et al. [29]. The OPS algorithms were applied using windows of 10 and increments of 5 variables where 100% of variables were tested and random cross-validation was applied and splits were set at 10% of the **X** matrix rows. In feedOPS, the convergence criteria were 2% as the minimum difference between two consecutive $RMSE_{CV}$ and 10 as the maximum number of loops. In iOPS, when the option to run the selection using feedOPS was used, the convergence criteria were the same as those used in feedOPS. The **X** matrix was divided into intervals of 10% of its size, limited in at least 50 variables. Additionally, the number of latent variables for OPS (hOPS) was calculated for each interval in iOPS.

The prediction set was used to validate the predictive capacity of the calibration models developed. Each property value from the laboratory typical analyses and the value predicted by the calibration models were compared. The root mean square error of prediction ($RMSE_P$) and cross-validation ($RMSE_{CV}$) and the correlation coefficient between the values measured in the typical laboratory analyses and those predicted by the model ($R_P$) and in the cross-validation ($R_{CV}$), were used as parameters to verify the predictive capacity of the calibration models.

Also, the CS constituents measured by typical laboratory analyses and those predicted by regression models were compared using the following regression model: $y = \beta_0 + \beta_1 \times x$, where x = predicted values; y = observed values; $\beta_0$ = intercept of equation; and $\beta_1$ = slope of equation. Regression was conducted according to the following statistical hypothesis [30]: $H_0$: $\beta_0 = 0$ and $\beta_1 = 1$; and $H_a$: not $H_0$. If the null hypothesis was not rejected, it was concluded that the equations accurately and precisely estimated the CS feeding value. When the regression coefficients β0 and β1 are 0 and 1, respectively, predicted and observed values were considered equivalent.

Estimates were also evaluated using the estimated value of the mean squared error of prediction (MSEP) and its components [31]: $MSEP = SB + MaF + MoF = 1/n \sum_{i=1} (x_i - y_i)^2$, $SB = (x_i - y_i)^2$, $Maf = (s_x - s_y)^2$, $Mof = 2s_x s_y (1 - r)$, where $x$ = predicted values; $y$ = observed

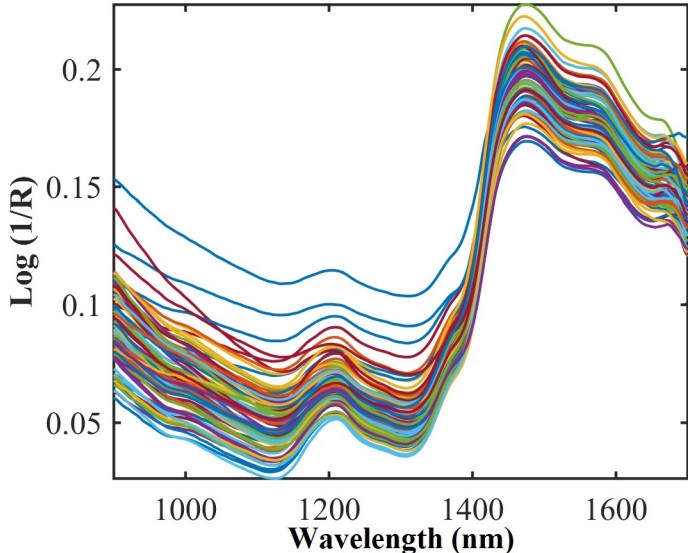

**Fig 1. NIR spectra of corn silage samples.**

**Table 1. Descriptive analysis of obtained values by standard methods of analysis and information about datasets used to perform calibration and prediction of regression models to predict the corn silage composition.**

| Property[1] | Mean | Minimum | Maximum | Standard deviation | Matrix size | | Pre-treatments[2] |
|---|---|---|---|---|---|---|---|
| | | | | | Calibration | Prediction | |
| DM | 30.19 | 20.43 | 41.61 | 4.142 | 58 × 256 | 25 × 256 | Norm + 2Der |
| DM at 105°C | 92.36 | 84.67 | 95.45 | 2.171 | 68 × 256 | 17 × 256 | 2Der + MC |
| OM | 95.58 | 86.26 | 97.71 | 1.710 | 62 × 256 | 23 × 256 | 2Der + MSC |
| CP | 6.87 | 5.00 | 9.06 | 0.743 | 64 × 256 | 16 × 256 | 1Der + Auto |
| NDIP | 0.87 | 0.45 | 1.68 | 0.251 | 66 × 256 | 16 × 256 | Auto + Smoth |
| ADIP | 0.28 | 0.12 | 0.64 | 0.080 | 68 × 256 | 18 × 256 | 1Der + Base |
| EE | 2.38 | 1.26 | 3.76 | 0.542 | 63 × 256 | 16 × 256 | 2Der + MC |
| NFC | 42.69 | 29.4 | 53.77 | 7.361 | 58 × 256 | 25 × 256 | 2Der + SNV |
| NDF | 45.91 | 34.15 | 57.81 | 5.689 | 65 × 256 | 16 × 256 | 2Der + Norm |
| apNDF | 43.73 | 32.92 | 56.33 | 5.852 | 63 × 256 | 16 × 256 | Smoth + 1Der + Auto |
| iNDF | 15.91 | 10.20 | 23.15 | 2.707 | 60 × 256 | 15 × 256 | 1Der + MC |
| ADF | 24.83 | 17.78 | 32.24 | 4.561 | 66 × 256 | 17 × 256 | 2Der + Base |
| apADF | 23.69 | 17.24 | 31.13 | 4.327 | 63 × 256 | 16 × 256 | 1Der + Auto |
| Lignin | 2.46 | 1.16 | 3.91 | 0.569 | 66 × 256 | 17 × 256 | 2Der |
| Starch | 25.07 | 11.00 | 40.53 | 7.197 | 57 × 256 | 24 × 256 | 2Der + Detrend |

[1] DM: % total dry matter; DM at 105°C: Dry matter in oven at 105°C (%DM); OM: Organic matter (%DM); CP: Crude protein (%DM); NDIP: Neutral detergent insoluble protein (%DM); ADIP: Acid detergent insoluble protein (%DM); EE: Ether extract (%DM); NFC: Non-fibrous carbohydrates (%DM); NDF: Neutral detergent insoluble fiber (%DM); apNDF: NDF corrected for ash and protein (%DM); iNDF: Indigestible NDF (%DM); ADF: Acid detergent insoluble fiber (% DM); and apADF: ADF corrected for ash and protein (%DM).

[2] Norm: Normalization; 2Der: Second derivative; MC: Mean center; MSC: Multiplicative scatter correction; 1Der: First derivative; Auto: Autoscale; Smooth: Smoothing; Base: Baseline; and SNV: Standard normal variate.

values; MSEP = mean squared error of prediction; SB = squared bias; MaF = component relative to the magnitude of random fluctuation; MoF = component relative to the model of random fluctuation; $s_X$ and $s_Y$ = standard deviations of predicted and observed values, respectively; and $r$ = Pearson's linear correlation between predicted and observed values. The smallest MSEP indicates the best model in the evaluation. These calculations can indicate

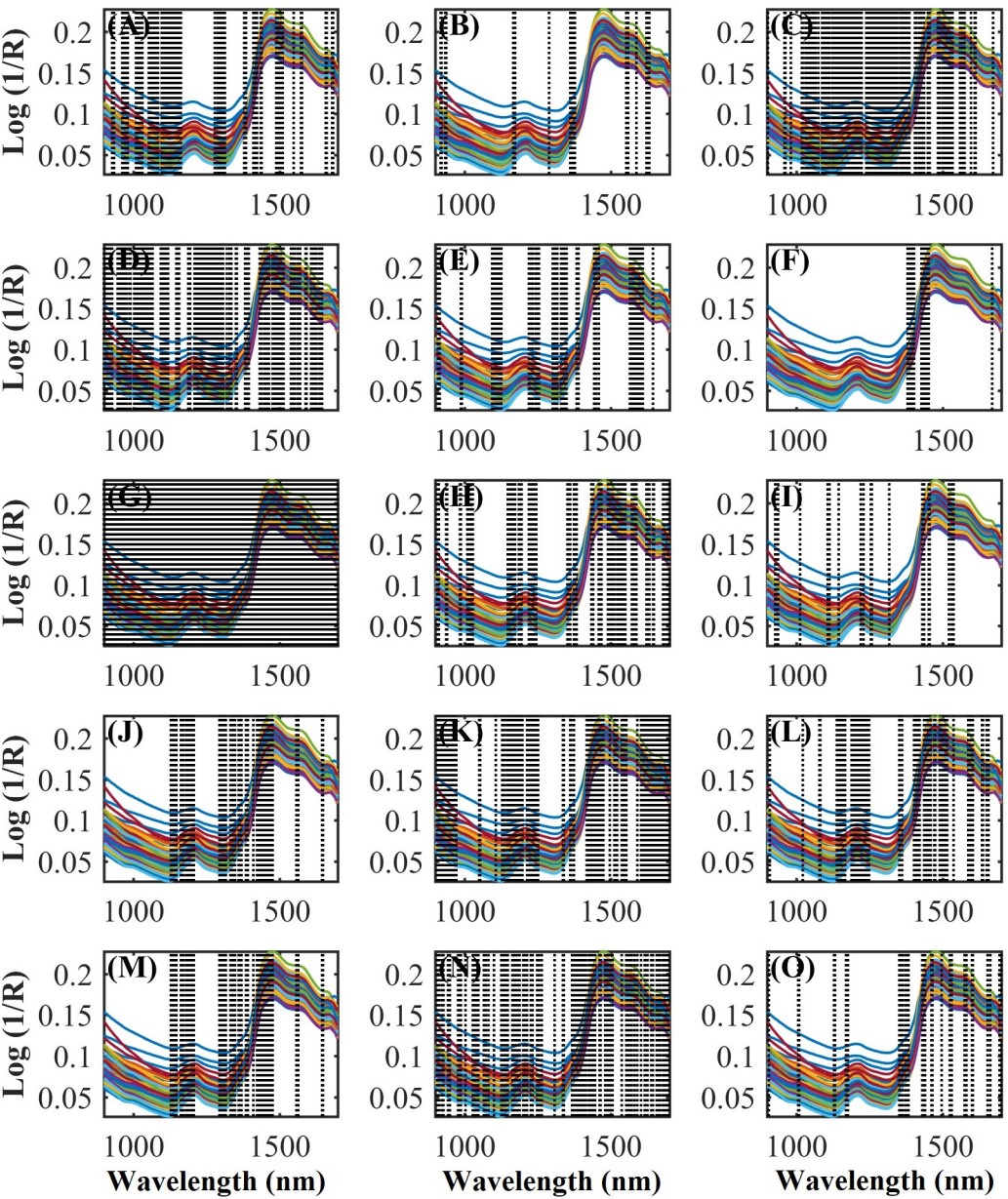

**Fig 2.** NIR spectrum regions selected by OPS algorithms in regression models building to predict the concentrations of (A) dry matter (DM), (B) dry matter at 105°C (DM105), (C) organic matter (OM), (D) crude protein (CP), (E) neutral detergent insoluble protein (NDIP), (F) acid detergent insoluble protein (ADIP), (G) ether extract (EE), (H) non-fibrous carbohydrates (NFC), (I) neutral detergent insoluble fiber (NDF), (J) NDF corrected for ash and protein (apNDF), (K) indigestible NDF (iNDF), (L) acid detergent insoluble fiber (ADF), (M) ADF corrected for ash and protein (FDAcp), (N) lignin and (O) starch of corn silage.

whether the model error is associated with the SB, errors related to the high dispersion of data around the mean (MaF), or systematic errors concerning the direction of the curve predicted (MoF).

For all variance and covariance calculations, the total number of observations was used as the denominator [31]. The prediction efficiency was determined by estimating the correlation and concordance coefficient (CCC) or reproducibility index as described by Tedeschi [32]. With the CCC, it is possible to ascertain whether the model is overestimating or underestimating the observed values (the closer to 1, the better the model) in addition to giving an indication of the model's precision and accuracy. Evaluation analyses were performed using the Model Evaluation System (32) and significance was established at $\alpha = 0.05$.

## Results and discussion

### CS composition

The NIR spectra of the CS samples (n = 94) are shown in Fig 1. The descriptive statistics of the size of the calibration and prediction sets, and the pre-treatments performed in the respective datasets for CS constituents are provided in Table 1. A wide variation in CS composition was found in the data set, which is commonly observed in production systems such as Brazil. The variation in concentrations of CP (5.00–9.06% MS), NDF (34.15–57.81% MS), and starch (11–40.53% MS) demonstrate the concerns of using book values for CS to formulate diets.

**Table 2. Performance parameters of developed PLS models to predict corn silage composition.**

| Models[2] | Items[1] | | | | | | | |
|---|---|---|---|---|---|---|---|---|
| | nvars | nlv | RMSE$_C$ | R$_C$ | RMSE$_{CV}$ | R$_{CV}$ | RMSE$_P$ | R$_P$ |
| DM | 85 | 4 | 2.703 | 0.72 | 3.184 | 0.59 | 2.683 | 0.74 |
| DM at 105˚C | 20 | 2 | 1.117 | 0.82 | 1.378 | 0.71 | 1.199 | 0.83 |
| OM | 134 | 9 | 0.291 | 0.95 | 0.633 | 0.76 | 0.824 | 0.64 |
| CP | 130 | 8 | 0.415 | 0.83 | 0.571 | 0.68 | 0.405 | 0.84 |
| NDIP | 60 | 8 | 0.116 | 0.84 | 0.131 | 0.8 | 0.136 | 0.76 |
| ADIP | 18 | 3 | 0.046 | 0.67 | 0.049 | 0.6 | 0.052 | 0.59 |
| EE | 256 | 8 | 0.263 | 0.87 | 0.461 | 0.55 | 0.289 | 0.86 |
| NFC | 86 | 7 | 1.691 | 0.96 | 2.208 | 0.93 | 2.092 | 0.92 |
| NDF | 20 | 7 | 1.438 | 0.97 | 1.788 | 0.95 | 1.473 | 0.97 |
| apNDF | 60 | 8 | 1.180 | 0.94 | 1.374 | 0.92 | 1.387 | 0.91 |
| iNDF | 117 | 9 | 0.952 | 0.92 | 1.423 | 0.81 | 1.211 | 0.87 |
| ADF | 70 | 7 | 1.113 | 0.95 | 1.615 | 0.90 | 1.318 | 0.94 |
| apADF | 60 | 8 | 1.180 | 0.94 | 1.346 | 0.92 | 1.387 | 0.91 |
| Lignin | 100 | 10 | 0.087 | 0.99 | 0.312 | 0.82 | 0.253 | 0.91 |
| Starch | 35 | 9 | 2.485 | 0.93 | 3.262 | 0.88 | 2.900 | 0.91 |

[1]nvars: Number of variables selected by OPS algorithms; nlv: Number of latent variables of the model; RMSE$_C$: Root mean square error of calibration; R$_C$: Correlation coefficient of calibration; RMSE$_{CV}$: Root mean square error of cross-validation; R$_{CV}$: Correlation coefficient of cross-validation; RMSE$_P$: Root mean square error of prediction; R$_P$: Correlation coefficient of prediction.

[2]DM: % total dry matter; DM at 105˚C: Dry matter in oven at 105˚C (%DM); OM: Organic matter (%DM); CP: Crude protein (%DM); NDIP: Neutral detergent insoluble protein (%DM); ADIP: Acid detergent insoluble protein (%DM); EE: Ether extract (%DM); NFC: Non-fibrous carbohydrates (%DM); NDF: Neutral detergent insoluble fiber (%DM); apNDF: NDF corrected for ash and protein (%DM); iNDF: Indigestible NDF (%DM); ADF: Acid detergent insoluble fiber (% DM); and apADF: ADF corrected for ash and protein (%DM).

For all CS constituents, the regression models obtained using pre-treated spectra resulted in lower RMSE$_{CV}$, indicating a better model fit and precision for regression models that underwent pre-treatments (Table 1). Pre-treatments are mathematical tools used to adjust multivariate regression models and correct random and systematic errors [33]. Pre-treatment could, for example, correct disturbances caused by physical phenomena such as

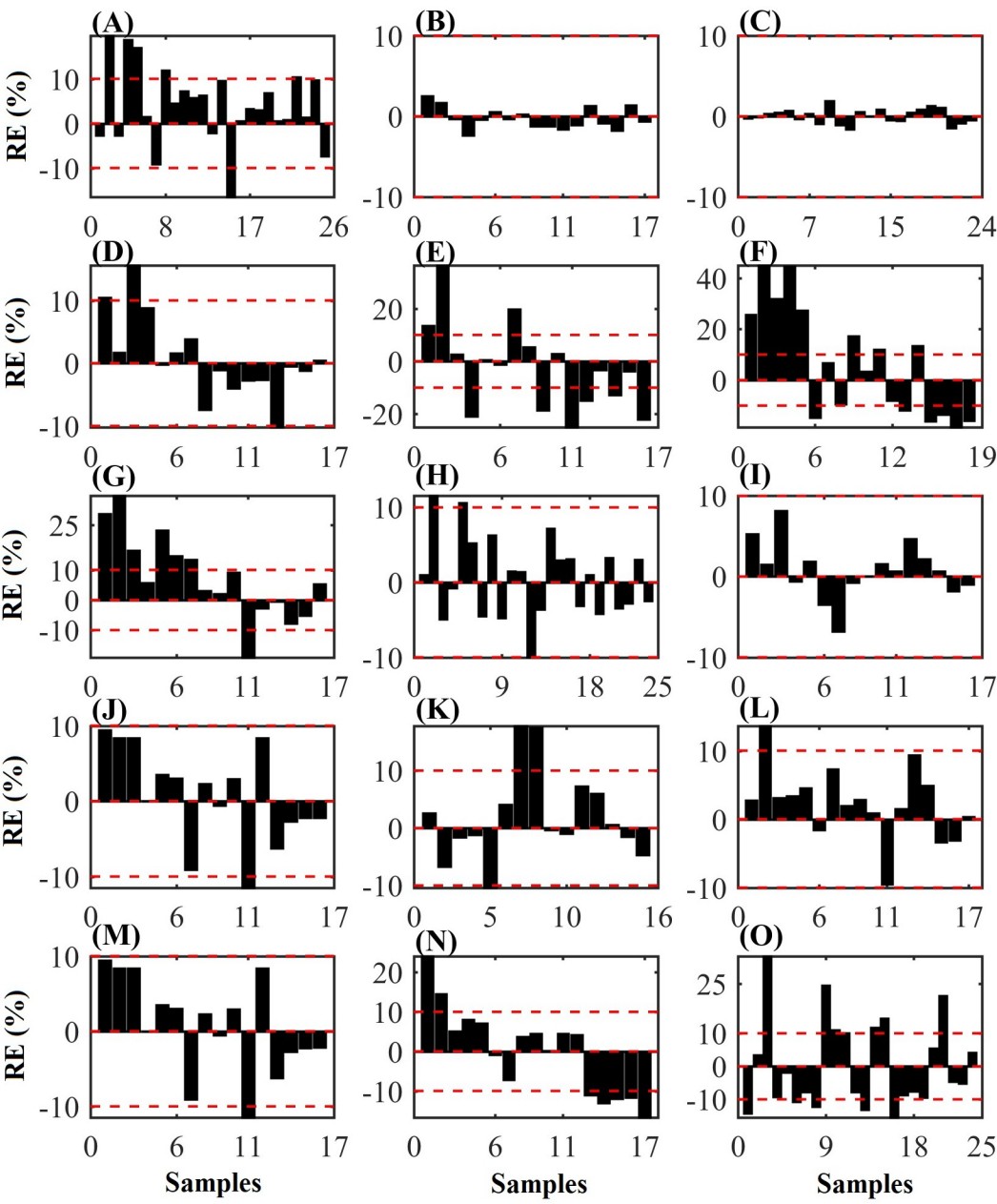

**Fig 3.** The relative error in the prediction of the concentrations of (A) dry matter (DM), (B) dry matter at 105°C (MS105), (C) organic matter (OM), (D) crude protein (CP), (E) neutral detergent insoluble protein (NDIP), (F) acid detergent insoluble protein (ADIP), (G) ether extract (EE), (H) non-fibrous carbohydrates (NFC), (I) neutral detergent insoluble fiber (NDF), (J) NDF corrected for ash and protein (apNDF), (K) indigestible NDF (iNDF), (L) acid detergent insoluble fiber (ADF), (M) ADF corrected for ash and protein (apADF), (N) lignin and (O) starch of corn silage by the regression models.

dispersions (multiplicative scatter correction), equalize the magnitude of samples through a normalization factor (normalization), and minimize interferences caused by noise (smoothing), among others. Therefore, the use of pre-treatments usually improves the goodness of fit of models.

The autoOPS, feedOPS, and iOPS algorithms were applied simultaneously during model development. Thus, it was possible to select regions of the NIR spectra with more relevant information and more correlated with the evaluated CS constituents (Fig 2). Variable selection is an important step in multivariate regression and has become a fundamental tool in many different research areas. Moreover, the proper choice of variables in the **X** matrix may improve

**Table 3.** Descriptive and validation statistics of the relationship among observed values by standard methods of analysis and predicted values by developed regression models.

| Item | DM[10] | | DM at 105°C[11] | | OM[12] | | EE[13] | | NFC[14] | |
|---|---|---|---|---|---|---|---|---|---|---|
| | Obs[15] | Pred[16] | Obs[15] | Pred[16] | Obs[15] | Pred[16] | Obs[15] | Pred[16] | Obs[15] | Pred[16] |
| Mean | 29 | 30.1 | 92.3 | 92.1 | 95.7 | 95.7 | 2.3 | 2.4 | 42.2 | 42.3 |
| SD[1] | 3.13 | 2.18 | 2.17 | 2.05 | 0.83 | 1.08 | 0.51 | 0.39 | 5.36 | 5.11 |
| Maximum | 37;2 | 33.6 | 95.3 | 96.2 | 97.4 | 97.7 | 3.1 | 3.2 | 51.3 | 52.9 |
| Minimum | 23;9 | 25.8 | 87.7 | 88.3 | 94.4 | 93.5 | 1.3 | 1.7 | 30.8 | 32.7 |
| $R^2$ | - | 0.535 | - | 0.677 | - | 0.383 | - | 0.716 | - | 0.918 |
| CCC[3] | - | 0.665 | - | 0.829 | - | 0.619 | - | 0.794 | - | 0.917 |
| Regression | | | | | | | | | | |
| Intercept | | | | | | | | | | |
| Estimate | - | -3.82 | - | 11.03 | - | 48.35 | - | -0.47 | - | 1.37 |
| SE[4] | - | 6.239 | - | 13.844 | - | 12.383 | - | 0.452 | - | 3.781 |
| Slope | | | | | | | | | | |
| Estimate | - | 1.1 | - | 0.88 | - | 0.49 | - | 1.14 | - | 0.96 |
| SE[4] | - | 0.205 | - | 0.15 | - | 0.129 | - | 0.183 | - | 0.089 |
| P-value[5] | - | 0.204 | - | 0.589 | - | 0.003 | - | 0.187 | - | 0.882 |
| MSEP[6] | - | 7.199 | - | 1.438 | | 0.678 | - | 0.084 | - | 4.377 |
| SB[7] | - | 0.87 | | 0.043 | - | 0.002 | - | 0.015 | - | 0.018 |
| Maf[8] | - | 0.059 | - | 0.055 | - | 0.285 | - | 0.003 | - | 0.032 |
| Mof[9] | - | 6.27 | - | 1.34 | - | 0.392 | - | 0.066 | - | 4.327 |

[1]Standard deviation.

[2]Determination coefficient.

[3]Correlation and concordance coefficient.

[4]Standard error;

[5]$H_0$: $\beta_0 = 0$ e $\beta_1 = 1$.

[6] Mean square error of prediction.

[7]Squared bias.

[8]Magnitude of random fluctuation.

[9]Random fluctuation of the model.

[10]Total dry matter.

[11]Dry matter in oven at 105°C (%DM).

[12]Organic matter (%DM).

[13]Ether extract (%DM).

[14]Non-fibrous carbohydrates (%DM).

[15]Observed values by standard methods of analysis.

[16]Predicted values by the developed regression model.

the goodness of fit of multivariate regression models. Thus, the proper selection of variables can be obtained by the triage of certain regions of the spectrum (wavelength set), which may minimize the error of prediction. As a result, more robust, simpler, and more accurate models may be obtained [34].

The number of variables selected by the OPS algorithms, the number of latent variables selected in the calibration process, and the performance parameters for all developed models are presented in Table 2. In general, models presented RMSE$_P$ values that suggest good fit of them. Those developed to estimate OM and ADIP presented a lower prediction correlation coefficient than the other models, demonstrating a possible fit problem for these models. The

**Table 4. Descriptive and validation statistics of the relationship among observed values by standard methods of analysis and predicted values by developed regression models.**

| Item | NDF[10] | | apNDF[11] | | iNDF[12] | | ADF[13] | | apADF[14] | |
|---|---|---|---|---|---|---|---|---|---|---|
| | Obs[15] | Pred[16] | Obs[15] | Pred[16] | Obs[15] | Pred[16] | Obs[15] | Pred[16] | Obs[15] | Pred[16] |
| Mean | 45.7 | 45.9 | 23.5 | 23.6 | 15.8 | 16.1 | 24.7 | 25.2 | 23.5 | 23.6 |
| SD[1] | 5.8 | 5.61 | 3.52 | 3.05 | 2.21 | 2.48 | 3.67 | 3.45 | 3.52 | 3.05 |
| Maximum | 55.5 | 55 | 30.2 | 29.6 | 20.6 | 19.7 | 31.6 | 31.7 | 30.2 | 29.6 |
| Minimum | 35.2 | 37.1 | 17.9 | 19.5 | 12.5 | 12.1 | 18.6 | 19.1 | 17.9 | 19.5 |
| R[2] | - | 0.929 | - | 0.825 | - | 0.743 | - | 0.874 | - | 0.825 |
| CCC[3] | - | 0.965 | - | 0.905 | - | 0.86 | - | 0.929 | - | 0.905 |
| Regression | | | | | | | | | | |
| Intercept | | | | | | | | | | |
| Estimate | - | -0.22 | - | -1.39 | - | 3.31 | - | -0.44 | - | -1.39 |
| SE[4] | - | 3.287 | - | 2.96 | - | 1.965 | - | 2.397 | - | 2.96 |
| Slope | | | | | | | | | | |
| Estimate | - | 1 | - | 1.06 | - | 0.78 | - | 1 | - | 1.06 |
| SE[4] | - | 0.071 | - | 0.125 | - | 0.12 | - | 0.094 | - | 0.125 |
| P-value[5] | - | 0.758 | - | 0.892 | - | 0.143 | - | 0.329 | - | 0.892 |
| MSEP[6] | | 2.168 | - | 1.925 | - | 1.466 | - | 1.737 | - | 1.925 |
| SB[7] | - | 0.084 | - | 0.004 | - | 0.091 | - | 0.239 | - | 0.004 |
| Maf[8] | - | <0.001 | - | 0.028 | - | 0.289 | - | <0.001 | - | 0.028 |
| Mof[9] | - | 2.084 | - | 1.893 | - | 1.087 | - | 1.498 | - | 1.893 |

[1]Standard deviation.

[2]Determination coefficient.

[3]Correlation and concordance coefficient.

[4]Standard error;

[5]$H_0$: $\beta_0 = 0$ e $\beta_1 = 1$.

[6] Mean square error of prediction.

[7]Squared bias.

[8]Magnitude of random fluctuation.

[9]Random fluctuation of the model.

[10]Neutral detergent insoluble fiber (%DM).

[11]NDF corrected for ash and protein (%DM).

[12]Indigestible NDF (%DM).

[13]Acid detergent insoluble fiber (%DM).

[14]ADF corrected for ash and protein (%DM).

[15]Observed values by standard methods of analysis.

[16]Predicted values by the developed regression model.

graphs with the relative prediction errors for each model are shown in Fig 3. These results indicate that the calibration models presented a good predictive capacity.

Zicarelli et al. [14] employed a NIR spectrophotometer to assess undried samples, yielding RP and RMSEP values of 0.9 and 1.9, respectively, for DM. These parameters are greater to those obtained in the current study. This disparity can be attributed to the fact that, in this study, the samples were dried and ground, which required the models to predict the water content, which is no longer present in the samples. The authors also provided model parameters for starch, CP, NDF, ADF, and EE, all of which exhibited lower RP and RMSEP values. The reduced RP reported by these authors suggests that the selection of variables can enhance model accuracy, given that their models were developed using a full-spectrum regression methodology. The lower RMSEP observed in their study can be attributed to the larger sample size they employed.

**Table 5. Descriptive and validation statistics of the relationship among observed values by standard methods of analysis and predicted values by developed regression models.**

| Item | CP[10] | | NDIP[11] | | ADIP[12] | | Lignin | | Starch | |
|---|---|---|---|---|---|---|---|---|---|---|
| | Obs[13] | Pred[14] | Obs[13] | Pred[14] | Obs[13] | Pred[14] | Obs[13] | Pred[14] | Obs[13] | Pred[14] |
| Mean | 6.8 | 6.8 | 0.8 | 0.8 | 0.3 | 0.3 | 2.4 | 2.4 | 24.4 | 24.1 |
| SD[1] | 0.76 | 0.62 | 0.21 | 0.15 | 0.07 | 0.04 | 0.53 | 0.35 | 7.12 | 6.27 |
| Maximum | 8.1 | 8.1 | 1.4 | 1.1 | 0.4 | 0.4 | 3.3 | 2.8 | 39.4 | 38.7 |
| Minimum | 5.2 | 5.8 | 0.5 | 0.5 | 0.1 | 0.2 | 1.3 | 1.6 | 12.6 | 11.5 |
| R[2] | - | 0.678 | - | 0.551 | - | 0.31 | - | 0.819 | - | 0.821 |
| CCC[3] | - | 0.819 | - | 0.713 | - | 0.546 | - | 0.831 | - | 0.903 |
| Regression | | | | | | | | | | |
| Intercept | | | | | | | | | | |
| Estimate | - | -0.17 | - | 0.02 | - | 0.03 | - | -0.89 | - | 0.57 |
| SE[4] | - | 1.23 | - | 0.188 | - | 0.084 | - | 0.394 | - | 2.495 |
| Slope | | | | | | | | | | |
| Estimate | - | 1.02 | - | 1.03 | - | 0.87 | - | 1.39 | - | 1.03 |
| SE[4] | - | 0.179 | - | 0.233 | - | 0.294 | - | 0.163 | - | 0.1 |
| P-value[5] | - | 0.966 | - | 0.536 | - | 0.735 | - | 0.07 | - | 0.893 |
| MSEP[6] | - | 0.164 | - | 0.019 | - | 0.003 | - | 0.064 | - | 8.412 |
| SB[7] | - | 0.001 | - | 0.002 | - | <0.001 | - | 0.002 | - | 0.046 |
| Maf[8] | - | <0.001 | - | <0.001 | - | <0.001 | - | 0.017 | - | 0.04 |
| Mof[9] | - | 0.163 | - | 0.017 | - | 0.003 | - | 0.045 | - | 8.326 |

[1]Standard deviation.

[2]Determination coefficient.

[3]Correlation and concordance coefficient.

[4]Standard error;

[5]$H_0$: $\beta_0 = 0$ e $\beta_1 = 1$.

[6] Mean square error of prediction.

[7]Squared bias.

[8]Magnitude of random fluctuation.

[9]Random fluctuation of the model.

[10]Crude protein (%DM).

[11]Neutral detergent insoluble protein (%DM).

[12]Acid detergent insoluble protein (%DM).

[13]Observed values by standard methods of analysis.

[14]Predicted values by the developed regression model.

The results of the independent dataset prediction for each CS constituent are presented in Tables 3–5. Corn silage concentrations of DM, DM at 105˚C, EE, CP, ADIP, NDIP, NDF, apNDF, iNDF, ADF, apADF, lignin, NFC, and starch were correctly estimated by the developed models as they did not reject the null hypothesis of intercept and slope equal to 0 and 1 respectively (P > 0.05). Moreover, the MSEP decomposition indicated that prediction errors

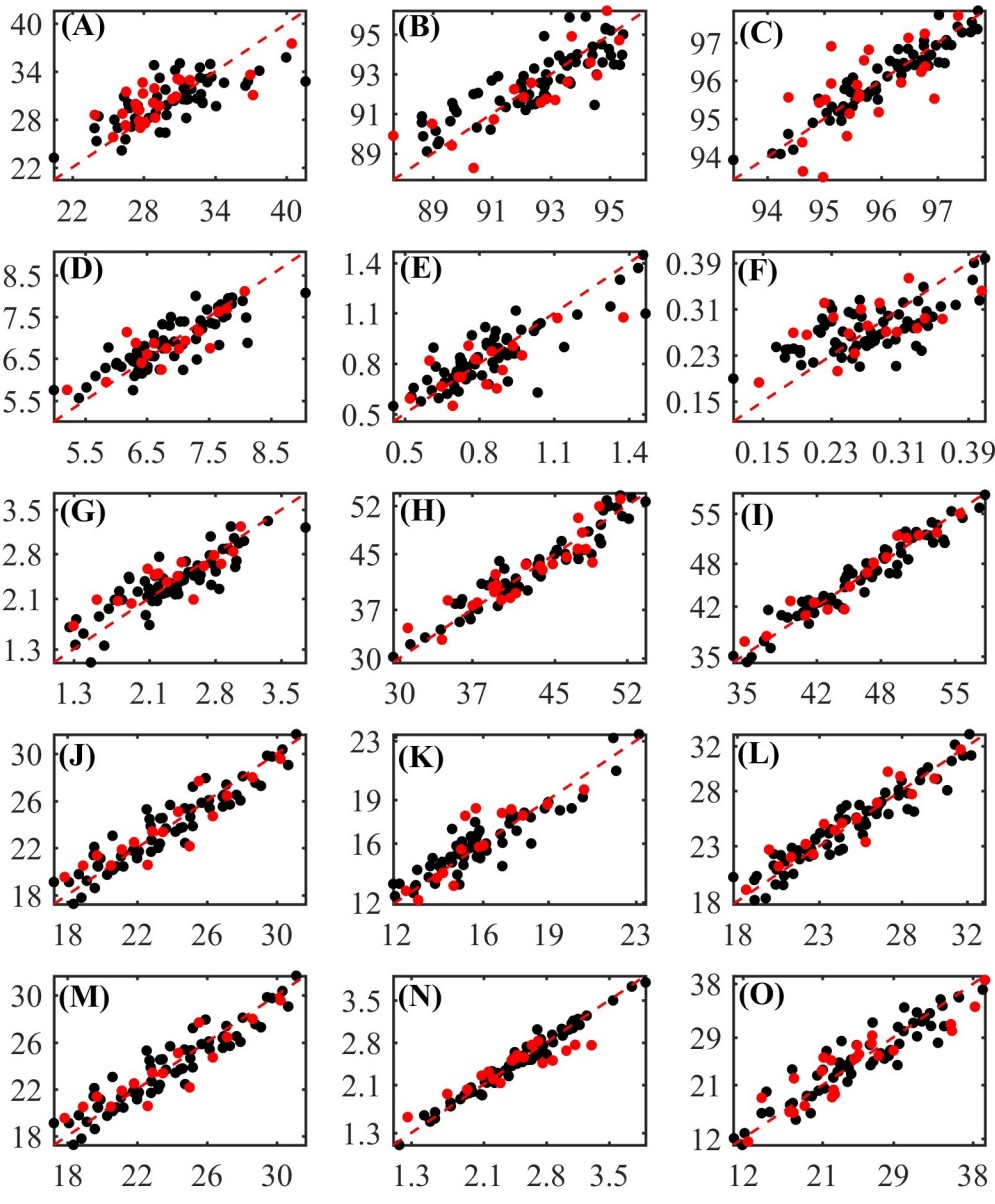

**Fig 4.** Measured (x axis) vs. predicted (y axis) values obtained by developed regression models for the concentrations of (A) dry matter (DM), (B) dry matter at 105˚C (MS105), (C) organic matter (OM), (D) crude protein (CP), (E) neutral detergent insoluble protein (NDIP), (F) acid detergent insoluble protein (ADIP), (G) ether extract (EE), (H) non-fibrous carbohydrates (NFC), (I) neutral detergent insoluble fiber (NDF), (J) NDF corrected for ash and protein (apNDF), (K) indigestible NDF (iNDF), (L) acid detergent insoluble fiber (ADF), (M) ADF corrected for ash and protein (apADF), (N) lignin and (O) starch of corn silage. (■) Calibration set; (●) Prediction set.

were mostly (more than 70% of the MSEP) associated with random errors (Mof), suggesting that the prediction errors were not associated with the developed models. Furthermore, the calibration models precisely estimated the concentrations of DM at 105°C, EE, CP, NDIP, NDF, apNDF, iNDF, ADF, apADF, lignin, NFC, and starch of CS, as the CCC was close to 1 (CCC $\geq$ 0.713); in contrast, models for predicting the concentrations of DM and ADIP presented CCC of 0.665 for DM and 0.546 for ADIP. The calibration model for predicting the OM content of CS had adjustment problems according to the regression analysis (P < 0.05), indicating poor accuracy. In addition, 42.23% of the MSEP of the prediction of OM concentration in CS was associated with the central tendency or bias (SB) and systematic errors (Maf), indicating a model adjustment problem.

Graphs with values measured by conventional laboratory methods and predicted by the regression models are shown in Fig 4.

The OM content is obtained by calculation (DM—Ash), thus cumulating the errors of the separate analyses, which may explain the lack of accuracy of the model. Moreover, the chemical constituents of OM can vary greatly across samples.

## Ruminal degradability parameters

The descriptive statistics of the size of the calibration set and the pre-treatments performed in the dataset for ruminal degradation parameters of CS are provided in Table 6. A wide variation in ruminal degradation parameters was detected in the study, demonstrating that the ruminal degradability of CS varies significantly. For example, CP ruminal degradation parameters ranged from 57.21 to 78.92 for fraction "a", 14.63 to 33.51 for fraction "b", and 0.03 to 0.08 for rate "kd". Average values are 60 for fraction "a", 24 for fraction "b" and 0.04 for "kd" [7]. This fact demonstrates the difficulty of using tabulated values and the need for

**Table 6. Descriptive analysis of obtained (%) by *in situ* technique and information about datasets used to perform calibration of regression models to predict the ruminal degradation parameters of corn silage by NIR spectra.**

| Parameter[1] | Mean | Minimum | Maximum | Standart deviation | Calibration matrix size | Pre-treatments |
|---|---|---|---|---|---|---|
| *Dry matter* | | | | | | |
| a | 48.81 | 37.48 | 58.47 | 5.548 | 22 × 256 | Smoth + 2Der |
| b | 39.63 | 33.91 | 48.59 | 3.936 | 21 × 256 | 1Der + Auto |
| kd | 0.02 | 0.01 | 0.03 | 0.004 | 21 × 256 | 1Der + Base |
| *Organic matter* | | | | | | |
| a | 48.11 | 36.32 | 58.22 | 5.823 | 21 × 256 | Smoth + 2Der |
| b* | 40.57 | 33.20 | 50.70 | 4.318 | - | - |
| kd* | 0.02 | 0.01 | 0.03 | 0.005 | - | - |
| *Crude protein* | | | | | | |
| A | 69.2 | 57.21 | 78.92 | 5.458 | 22 × 256 | 1Der + Norm |
| b* | 23.49 | 14.63 | 33.51 | 4.505 | - | - |
| kd | 0.05 | 0.03 | 0.08 | 0.014 | 21 × 256 | 2Der + MC |
| *Neutral detergent fiber* | | | | | | |
| b | 62.08 | 49.13 | 74.6 | 7.325 | 21 × 256 | Detrend + MC |
| I | 33.27 | 23.57 | 42.27 | 5.704 | 21 × 256 | Auto + SNV |
| kd | 0.03 | 0.02 | 0.05 | 0.006 | 21 × 256 | 2Der + MC |

[1]a: Readily soluble fraction (%); b: Potentially degradable fraction in the rumen (%); kd: Rate constant for degradation of b (per h); I: Undegradable fraction.

*Could not find a suitable model.

faster and more viable analytical methods, which will allow for greater precision in the formulation of diets.

For all ruminal degradability parameters, the regression models obtained using pre-treated spectra resulted in lower values of $RMSE_{CV}$, indicating a better model fit and precision. It was not possible to find suitable models to predict the fraction "b" of OM and CP, and the "kd" of OM. The microbial contamination of residual particles of incubated feeds is an important source of errors in the *in situ* method, resulting in the underestimation of CP degradability [18,35,36]. Thus, such a source of error can impact the construction of prediction models for CP and OM degradation parameters.

The selected regions are shown in Fig 5. The number of variables selected by the OPS algorithms, the number of latent variables selected in the calibration process, and the performance parameters for all developed models are presented in Table 7. The developed models to predict "a" fraction of DM, OM and CP achieved $R_{cv}$ values between 0.802 to 0.985 and the model of "a" fraction of CP showed the highest $RMSE_{cv}$ among them. The model to predict the "b"

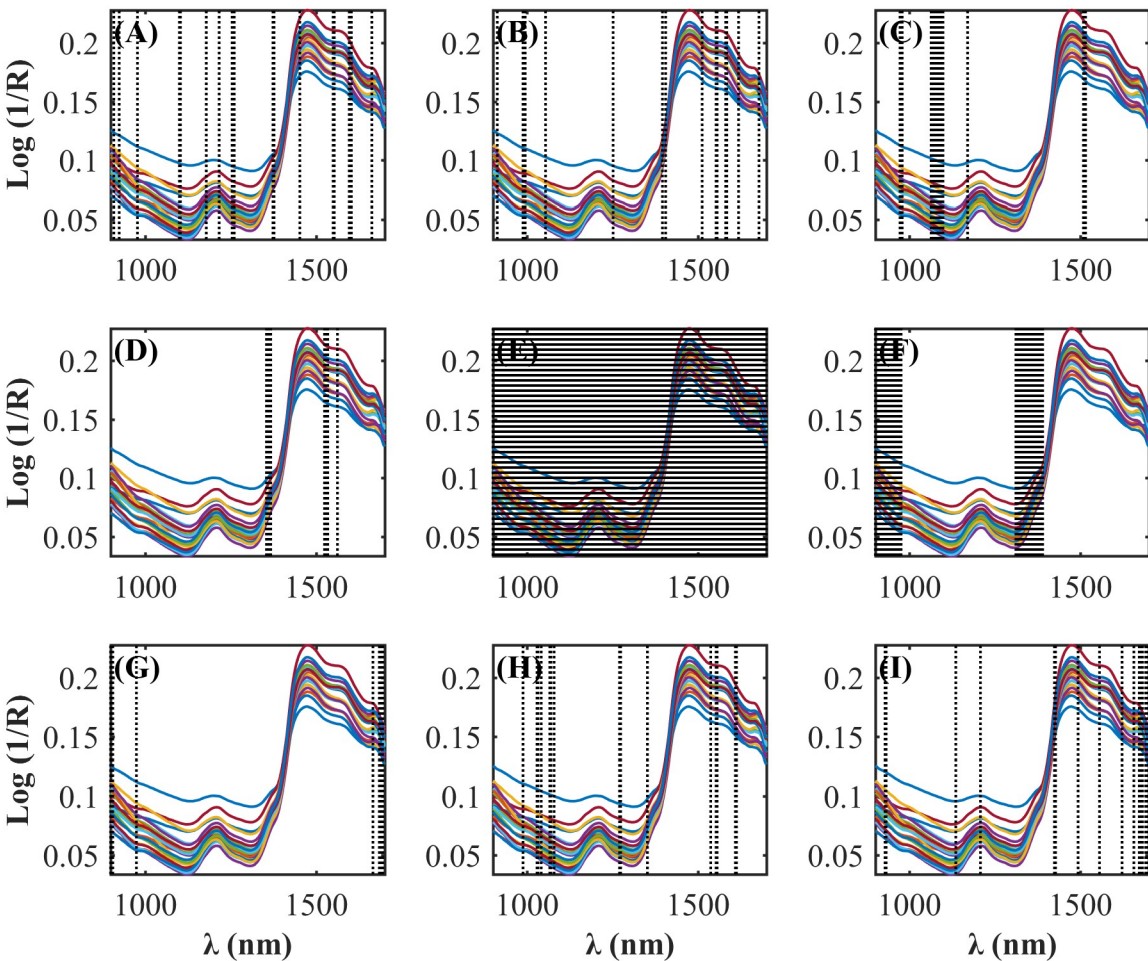

**Fig 5.** NIR spectrum regions selected by OPS algorithms in regression models building to predict the readily soluble fraction "a" of **(A)** dry matter (DM), **(B)** organic matter (OM), **(C)** crude protein (CP), the potentially degradable fraction in the rumen "b" **(D)** of DM, **(E)** of neutral detergent fiber (NDF), **(F)** NDF undegradable fraction "I", rate constant for degradation of b "kd" of **(G)** DM, **(H)** CP, and **(I)** NDF of corn silage.

**Table 7. Performance parameters of developed PLS models to predict ruminal degradation parameters of corn silage by NIR spectra.**

| Models[2] | Items[1] | | | | | |
|---|---|---|---|---|---|---|
| | nvars | nlv | RMSE$_C$ | R$_C$ | RMSE$_{CV}$ | R$_{CV}$ |
| *Dry matter* | | | | | | |
| A | 19 | 8 | 0.559 | 0.995 | 1.468 | 0.964 |
| B | 10 | 4 | 1.459 | 0.903 | 2.146 | 0.776 |
| Kd | 8 | 6 | 0.001 | 0.959 | 0.002 | 0.915 |
| *Organic matter* | | | | | | |
| A | 15 | 8 | 0.578 | 0.995 | 0.999 | 0.985 |
| b* | - | - | - | - | - | - |
| kd* | - | - | - | - | - | - |
| *Crude protein* | | | | | | |
| A | 15 | 7 | 1.538 | 0.958 | 3.237 | 0.802 |
| b* | - | - | - | - | - | - |
| kd | 15 | 2 | 0.006 | 0.913 | 0.009 | 0.758 |
| Neutral detergent fiber | | | | | | |
| b | 256 | 10 | 0.812 | 0.994 | 5.535 | 0.655 |
| I | 51 | 1 | 4.103 | 0.69 | 4.829 | 0.530 |
| kd | 16 | 10 | 0.001 | 0.991 | 0.003 | 0.904 |

[1]nvars: Number of variables selected by OPS algorithms; nlv: Number of latent variables of the model; RMSE$_C$: Root mean square error of calibration; R$_C$: Correlation coefficient of calibration; RMSE$_{CV}$: Root mean square error of cross-validation; R$_{CV}$: Correlation coefficient of cross-validation.

[2] a: Readily soluble fraction (%); b: Potentially degradable fraction in the rumen (%); kd: Rate constant for degradation of b (per h); I: Undegradable fraction.

*Could not find a suitable model.

fraction of DM presented a R$_{cv}$ of 0.776 and RMSE$_{cv}$ of 2.146. The variable selection was not efficient to improve the model to predict the fraction "b" of the NDF, since all the variables of the spectrum were used. The prediction models for the NDF "b" and "I" fractions, and "kd" rate presented Rcv of 0.66, 0.53, and 0.90.

Graphs illustrating the relationship between values measured by the *in situ* method and predicted by the regression models are shown in Fig 6. The development of more comprehensive regression models would increase their accuracy and precision, but this would require the addition of corn silage samples collected across multiple years and various environments; this would increase the spectral, composition, and degradability diversity of the samples compared to those presented in the current study.

Thus, this study has demonstrated that NIR spectroscopy associated with chemometric methods has the potential to be used in the prediction of more complex variables, facilitating the correct and precise application of these variables in the field. Thus, new and broader research in this area should be encouraged.

## Conclusion

The regression models developed to predict CS composition accurately estimated the concentrations of DM, DM at 105°C, CP, ADIP, NDIP, EE, NDF, apNDF, iNDF, ADF, apADF, lignin, NFC, and starch. The models developed to predict the ruminal degradation parameters showed moderate predictive performance and has potential to be used by nutritionists as a rapid prediction tool in the field. Further development of the models using larger and more diverse sample datasets would improve model robustness and accuracy.

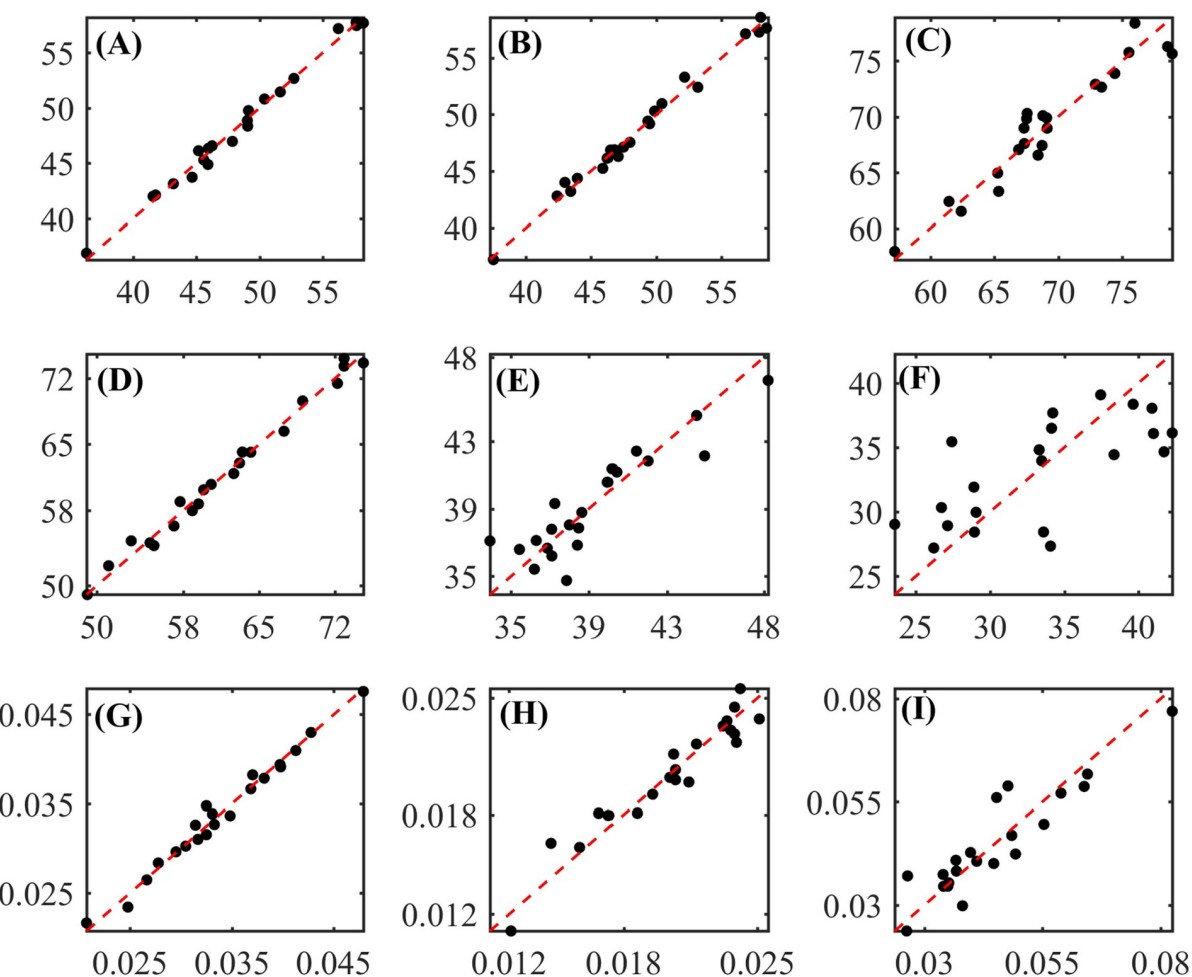

**Fig 6.** Measured (x axis) vs. predicted (y axis) values obtained by developed regression models for the readily soluble fraction "a" of **(A)** dry matter (DM), **(B)** organic matter (OM), **(C)** crude protein (CP), the potentially degradable fraction in the rumen "b" **(D)** of DM, **(E)** of neutral detergent fiber (NDF), **(F)** NDF undegradable fraction "I", rate constant for degradation of b "kd" of **(G)** DM, **(H)** CP, and **(I)** NDF of corn silage.

## Supporting information

**S1 File. Near-infrared spectra and laboratory values used to develop regression models for predicting composition of corn silage.**
(XLSX)

**S2 File. Near-infrared spectra and ruminal *in-situ* degradation parameters used to develop models for predicting ruminal degradability of corn silage.**
(XLSX)

## Author Contributions

**Conceptualization:** Sebastião de Campos Valadares Filho.

**Data curation:** Pauliane Pucetti, Sebastião de Campos Valadares Filho.

**Formal analysis:** Pauliane Pucetti.

**Funding acquisition:** Sebastião de Campos Valadares Filho.

**Investigation:** Pauliane Pucetti, Sebastião de Campos Valadares Filho, Jussara Valente Roque, Julia Travassos da Silva, Kellen Ribeiro de Oliveira, Flavia Adriane Sales Silva, Wilson Junior Cardoso, Fabyano Fonseca e Silva.

**Methodology:** Pauliane Pucetti, Sebastião de Campos Valadares Filho, Jussara Valente Roque, Julia Travassos da Silva, Kellen Ribeiro de Oliveira, Flavia Adriane Sales Silva, Wilson Junior Cardoso, Fabyano Fonseca e Silva.

**Project administration:** Sebastião de Campos Valadares Filho.

**Resources:** Sebastião de Campos Valadares Filho.

**Software:** Pauliane Pucetti, Jussara Valente Roque, Julia Travassos da Silva, Kellen Ribeiro de Oliveira, Flavia Adriane Sales Silva, Wilson Junior Cardoso, Fabyano Fonseca e Silva.

**Supervision:** Sebastião de Campos Valadares Filho.

**Validation:** Pauliane Pucetti, Sebastião de Campos Valadares Filho, Jussara Valente Roque, Julia Travassos da Silva, Kellen Ribeiro de Oliveira, Flavia Adriane Sales Silva, Wilson Junior Cardoso, Fabyano Fonseca e Silva.

**Visualization:** Pauliane Pucetti, Sebastião de Campos Valadares Filho, Jussara Valente Roque, Julia Travassos da Silva, Kellen Ribeiro de Oliveira, Flavia Adriane Sales Silva, Wilson Junior Cardoso, Fabyano Fonseca e Silva, Kendall Carl Swanson.

**Writing – original draft:** Pauliane Pucetti.

**Writing – review & editing:** Pauliane Pucetti, Sebastião de Campos Valadares Filho, Jussara Valente Roque, Julia Travassos da Silva, Kellen Ribeiro de Oliveira, Flavia Adriane Sales Silva, Wilson Junior Cardoso, Fabyano Fonseca e Silva, Kendall Carl Swanson.

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
