## [Decision Letter · Decision Letter 0]

1 Sep 2023

PONE-D-23-18553Predicting ruminal degradability and chemical composition of corn silage using near-infrared spectroscopy and multivariate regressionPLOS ONE

Dear Dr. Pucetti,

Thank you for submitting your manuscript to PLOS ONE. After careful consideration, we feel that it has merit but does not fully meet PLOS ONE’s publication criteria as it currently stands. Therefore, we invite you to submit a revised version of the manuscript that addresses the points raised during the review process.

We look forward to receiving your revised manuscript.

Kind regards,

Aziz ur Rahman Muhammad

Academic Editor

PLOS ONE

Journal Requirements:

"This work was supported by the National Council of Scientific and Technological Development (CNPq - Grant Number -465377/2014-9), National Institute of Science and Technology in Animal Science (INCT – Ciência Animal), and Coordination of Improvement of Personal Higher Education (CAPES)."

Additional Editor Comments:

Dear Authors,

I would like to invite authors to revise the manuscript as suggested by reviewer before final publication of the manuscript.

Reviewers' comments:

Reviewer's Responses to Questions

**Comments to the Author**

1. Is the manuscript technically sound, and do the data support the conclusions?

Reviewer #1: Yes

Reviewer #2: Yes

Reviewer #3: Yes

2. Has the statistical analysis been performed appropriately and rigorously? 

Reviewer #1: Yes

Reviewer #2: Yes

Reviewer #3: Yes

3. Have the authors made all data underlying the findings in their manuscript fully available?

Reviewer #1: Yes

Reviewer #2: Yes

Reviewer #3: Yes

4. Is the manuscript presented in an intelligible fashion and written in standard English?

Reviewer #1: Yes

Reviewer #2: Yes

Reviewer #3: No

5. Review Comments to the Author

Reviewer #1: The paper addresses an interesting issue, the use of NIR, which represents an interesting, low-impact and very rapid precision technology. Indeed, I do not find correct to use the validation of in vivo degradability with a very low number of samples (about 20).

I suggest to modify the paper without the in vivo test. The authors could divide it into 2 different papers.

Other important issues:

Line 80- Sample collection and preparation

Add information concerning:

-how much samples have been collected?

-time of collection

-which corn hybrid

-climate characteristics of the samples site

Line 88-I don’t understand the numbers of analyzed samples. 94 samples were analyzed with NIRS but only 23 were tested in vivo. Please explain that.

I suggest to compare the NIR spectra presented in this paper with that one published in the article Zicarelli, F.; et al.Agronomy 2023, 13, 634. https://doi.org/10.3390/ agronomy13030634

Reviewer #2: The authors have performed the analysis in a scientific manner and presented it in a wholely defined manner. I highly recommend the acceptance of the manuscript with minor revision

1. Kennard-stone algorithm is utilized for dataset classification. Please add few advantages and limitations of the algorithm.

2. Please explain concordance co-efficient.

I have no major issues prior submission to the manuscript except above stated comments.

Reviewer #3: General

This manuscript needs some revision before suitable for publication and the overall structure needs to be revised. The authors do need to provide further justification of the study as some similar studies have been published but the authors have not cited these in the introduction and identified the new information provided by their work. The novel aspects of the work need to be clearly identified.

The number of abbreviations in the manuscript is excessive and reduces the overall readability. The number of abbreviations needs to be reduced.

A combined results and discussion section can be difficult to read, I would suggest that the results and discussions sections are separated.

Line 23: delete additional ‘the’

Line 25 - 26: This needs to be rephrased. I presume the 95 samples were for development and the 23 samples were for validation? The sentence should reflect this, currently states development for both.

Line 27: What are the ‘typical analytical methods’?

Line 94: Change ‘7’ to ‘seven’.

Line 101: Why was this the considering the appropriate forage: concentrate ratio and diet composition?

Line 344: It would be better to use different shapes as colour version will be online only, consider readers that may print the manuscript in black and white.

Line 392: Replace the word ’globular’ with an appropriate word.

Table 1: Be consistent with the number of decimal places that data is presented to. Error should be presented as one more decimal place than mean (majority are okay but some values need to be corrected).

6. PLOS authors have the option to publish the peer review history of their article (what does this mean?). If published, this will include your full peer review and any attached files.

Reviewer #1: No

Reviewer #2: **Yes: **Tulika Bhardwaj

Reviewer #3: No

---

## [Author Response · Author response to Decision Letter 0]

11 Oct 2023

October 10th, 2023 – Fargo, ND, USA

Manuscript Number: PONE-D-23-18553

Title: Predicting ruminal degradability and chemical composition of corn silage using near-infrared spectroscopy and multivariate regression

Dear Section Editor Aziz ur Rahman Muhammad,

 We are submitting a revised version of the manuscript number PONE-D-23-18553 “Predicting ruminal degradability and chemical composition of corn silage using near-infrared spectroscopy and multivariate regression”. We would like to thank you for the opportunity to revise and improve the manuscript. Also, we thank the editor and reviewers for the important contributions to our manuscript.

 We have tried to incorporate the suggestions as much as we could. All modifications and replies of questions or concerns are detailed described below. In the revised manuscript, the changes are in green (reviewer #1), red (reviewer #2), and blue (reviewer #3) colors to facilitate the revision process.

 Please, let us know if you have any additional comments or criticism. We look forward to your positive feedback on this revised version.

Thanks, in advance!

Sincerely,

The authors. 

Replies to Reviewer #1

Reviewer #1: The paper addresses an interesting issue, the use of NIR, which represents an interesting, low-impact and very rapid precision technology. Indeed, I do not find correct to use the validation of in vivo degradability with a very low number of samples (about 20).

I suggest to modify the paper without the in vivo test. The authors could divide it into 2 different papers.

AU: Thank you for your careful and constructive criticisms regarding our manuscript. We appreciate your thoughtful consideration of our work. Regarding the in vivo degradability validation, we would like to clarify that our study examined in situ degradability evaluation, which is the most common approach for measuring nutrient degradability by researchers. Our objective was to explore the possibility of developing predictive models given the inherent challenges in obtaining in situ degradability parameters, including the need for cannulated animals and the substantial number of samples to be processed and analyzed. Despite the limited sample size, we implemented cross-validation to enhance the reliability of our findings. In this manner, even with the constrained number of samples, we can gain insights into the feasibility of obtaining in situ degradability parameters using NIR spectroscopy. This insight could guide future studies. We had pointed out in the discussion and conclusion that larger experiments are needed to improve the prediction models.

We believe that presenting both the in situ evaluation and chemical composition analysis in a single paper allows for a more comprehensive nutritional assessment of using NIR spectroscopy for evaluating corn silage. 

Other important issues:

Line 80- Sample collection and preparation

Add information concerning:

-how much samples have been collected?

-time of collection

-which corn hybrid

-climate characteristics of the samples site

AU: We collected a total of 94 samples, all of which were evaluated. However, the specifics you've mentioned, such as the exact time of collection, corn hybrid details, and precise climate characteristics, unfortunately, aren't available due to the nature of the sample collection process, which involved collaboration with rural producers and technical consultants, who did not have precise control over these factors during sample collection. Also, our objectives were not to examine different hybrids, but were to examine feed stuffs with different nutrient composition and degradability. We appreciate your understanding and remain available for any further inquiries.

Line 88-I don’t understand the numbers of analyzed samples. 94 samples were analyzed with NIRS but only 23 were tested in vivo. Please explain that.

AU: Thank you for your question regarding the number of analyzed samples. We indeed assessed all 94 samples for their chemical composition using NIRS. However, only 23 of them were subjected to in situ degradability assessment. This limitation can be attributed to various factors. Firstly, the quantity of samples provided by our collaborators was constrained, as in situ evaluation requires a larger sample volume. Additionally, our ability to evaluate more samples was restricted due to resource limitations at that time. These limitations included the availability of cannulated animals and the labor required to conduct the in situ assays. We hope this provides clarity, and we appreciate your valuable inquiry.

I suggest to compare the NIR spectra presented in this paper with that one published in the article Zicarelli, F.; et al. Agronomy 2023, 13, 634. https://doi.org/10.3390/ agronomy13030634

AU: Thank you for your suggestion. However, it appears that the referenced paper does not provide the NIR spectra used in their study. Instead, we utilized the prediction parameters from the models presented in the article you suggested, as the NIR spectra data was not available. Please, check them on lines 309-318. If there are any additional sources or data you recommend for comparison, we would be more than happy to consider them.

Replies to Reviewer #2

Reviewer #2: The authors have performed the analysis in a scientific manner and presented it in a wholely defined manner. I highly recommend the acceptance of the manuscript with minor revision.

AU: Thank you very much for your positive evaluation and recommendations. We are grateful for your encouraging feedback and recommendation for the acceptance of the manuscript with minor revisions. Your insights have been invaluable.

1. Kennard-stone algorithm is utilized for dataset classification. Please add few advantages and limitations of the algorithm.

Thank you for the valuable suggestion. We have now included a description of the Kennard-Stone algorithm in the Methods section of the paper, outlining its advantages. Specifically, we have included information on the advantages of the Kennard-Stone algorithm, which can be found in lines 188-192 of the revised manuscript. 

2. Please explain concordance co-efficient.

We have described in the text the meaning of the CCC and the reference for the reader to go if they want more information.

I have no major issues prior submission to the manuscript except above stated comments.

Replies to Reviewer #3

Reviewer #3: General

This manuscript needs some revision before suitable for publication and the overall structure needs to be revised. The authors do need to provide further justification of the study as some similar studies have been published but the authors have not cited these in the introduction and identified the new information provided by their work. The novel aspects of the work need to be clearly identified.

The number of abbreviations in the manuscript is excessive and reduces the overall readability. The number of abbreviations needs to be reduced.

A combined results and discussion section can be difficult to read, I would suggest that the results and discussions sections are separated.

AU: Thank you for your valuable feedback and constructive suggestions. We greatly appreciate your time and effort in reviewing our manuscript. Regarding the introduction, we agree with your comments. We have revised the manuscript and incorporated additional information to provide further justification for our study. Please refer to the introduction, specifically lines 72-76. Additionally, we've also made adjustments to the objectives of the study to ensure alignment with your suggestions. The updated objectives can be found in lines 79 and 80.

In response to the concern about the excessive use of abbreviations, we understand the importance of readability. However, the majority of the abbreviations are commonly used in the literature, which results in a more efficient presentation of the material.

Regarding the structure of the manuscript, we have carefully considered your suggestion to separate the results and discussion sections. However, we believe that a combined results and discussion section best serves the purpose of this article, providing a seamless flow of information and aiding in the visualization of data. We believe that in this case separating the results and discussion would result in too much repetition of the results.

Once again, we appreciate your thorough review and will make sure to address each concern diligently.

Line 23: delete additional ‘the’

AU: Thanks for your suggestion. 

Line 25 - 26: This needs to be rephrased. I presume the 95 samples were for development and the 23 samples were for validation? The sentence should reflect this, currently states development for both.

AU: Thank you for your suggestion. We have rephrased the respective sentences to accurately reflect the sample allocation. Out of a total of 94 collected samples, 94 were utilized for developing the composition models. Within this dataset, a subset of 23 samples was specifically used for the in situ degradability assessment. Please check lines 24-28.

Line 27: What are the ‘typical analytical methods’?

AU: Thank you for pointing that out. In lines 26-27, I have rephrased it to “wet chemistry methods”, that refers to analytical techniques that are routinely utilized in animal nutrition laboratories. These methods are commonly employed for sample analysis and are recognized as standard practices in the field, ensuring accurate assessment of food composition and other relevant parameters.

Line 94: Change ‘7’ to ‘seven’.

AU: “7” was replaced by “seven”. The sentence now reads (Line 100).

Line 101: Why was this the considering the appropriate forage: concentrate ratio and diet composition?

Thank you for your valuable feedback regarding forage:concentrate ratio. We have made the necessary modifications to the text to address your question. The revised text now reads (Lines 106-111):

"The bulls were fed a 70:30 (DM basis) corn silage:concentrate (corn, soybean meal, wheat bran, urea, ammonium sulfate, salt, bicarbonate, and mineral) diet for ad libitum intake. The diet contained 12% crude protein (DM basis). The forage-to-concentrate ratio used was based on providing a balanced diet with diverse feed composition with enough fiber and energy to not compromise microbial growth. The animals were adapted to the experimental diet and conditions for 30 days before the in situ incubations."

Line 344: It would be better to use different shapes as colour version will be online only, consider readers that may print the manuscript in black and white.

Thank you for your suggestion. We appreciate your consideration for readers who may print the manuscript in black and white. We have modified the shapes in the figure and its accompanying legend to ensure clarity and differentiation even in black and white format. The sentence now reads (Line 359-360).

Line 392: Replace the word ’globular’ with an appropriate word.

AU: “globular” was replaced by “comprehensive”. The sentence now reads (Line 409).

Table 1: Be consistent with the number of decimal places that data is presented to. Error should be presented as one more decimal place than mean (majority are okay but some values need to be corrected).

Thank you for your I have revised the table to ensure consistency in the number of decimal places for the data. The error values now align with your suggestion of presenting one more decimal place than the mean values.

---

## [Decision Letter · Decision Letter 1]

30 Oct 2023

PONE-D-23-18553R1Predicting ruminal degradability and chemical composition of corn silage using near-infrared spectroscopy and multivariate regressionPLOS ONE

Dear Dr. Pucetti,

Thank you for submitting your manuscript to PLOS ONE. After careful consideration, we feel that it has merit but does not fully meet PLOS ONE’s publication criteria as it currently stands. Therefore, we invite you to submit a revised version of the manuscript that addresses the points raised below. Please submit your revised manuscript by Dec 14 2023 11:59PM. If you will need more time than this to complete your revisions, please reply to this message or contact the journal office at plosone@plos.org. Please include the following items when submitting your revised manuscript:A rebuttal letter that responds to each point raised by the academic editor and reviewer(s). You should upload this letter as a separate file labeled 'Response to Reviewers'.A marked-up copy of your manuscript that highlights changes made to the original version. You should upload this as a separate file labeled 'Revised Manuscript with Track Changes'.An unmarked version of your revised paper without tracked changes. You should upload this as a separate file labeled 'Manuscript'.If applicable, we recommend that you deposit your laboratory protocols in protocols.io to enhance the reproducibility of your results. Protocols.io assigns your protocol its own identifier (DOI) so that it can be cited independently in the future. For instructions see: https://journals.plos.org/plosone/s/submission-guidelines#loc-laboratory-protocols. Additionally, PLOS ONE offers an option for publishing peer-reviewed Lab Protocol articles, which describe protocols hosted on protocols.io. Read more information on sharing protocols at https://plos.org/protocols?utm_medium=editorial-email&utm_source=authorletters&utm_campaign=protocols.

We look forward to receiving your revised manuscript.

Kind regards,

Aziz ur Rahman Muhammad

Academic Editor

PLOS ONE

Journal Requirements:

Additional Editor Comments:

Dear Authors

Thanks for revising the current manuscript according to reviewer suggestion. However, you reported that you have provided the revised version with changes that are in different color but in the revised manuscript i am unable to find different color. Therefore, i request author to resubmit the revised file indicating changes so that i could finalize the current manuscript.

Regards

Reviewers' comments:

Reviewer's Responses to Questions

**Comments to the Author**

1. If the authors have adequately addressed your comments raised in a previous round of review and you feel that this manuscript is now acceptable for publication, you may indicate that here to bypass the “Comments to the Author” section, enter your conflict of interest statement in the “Confidential to Editor” section, and submit your "Accept" recommendation.

Reviewer #3: All comments have been addressed

2. Is the manuscript technically sound, and do the data support the conclusions?

Reviewer #3: Yes

3. Has the statistical analysis been performed appropriately and rigorously? 

Reviewer #3: Yes

4. Have the authors made all data underlying the findings in their manuscript fully available?

Reviewer #3: Yes

5. Is the manuscript presented in an intelligible fashion and written in standard English?

Reviewer #3: Yes

6. Review Comments to the Author

Reviewer #3: (No Response)

7. PLOS authors have the option to publish the peer review history of their article (what does this mean?). If published, this will include your full peer review and any attached files.

Reviewer #3: No

---

## [Author Response · Author response to Decision Letter 1]

7 Dec 2023

I have incorporated the requested changes in the "Revised Manuscript with Track Changes." All modifications have been indicated in the "Response to Reviewers" document, distinctly highlighted using Text Highlight Colors for clarity.

---

## [Editor Report · Decision Letter 2]

14 Dec 2023

Predicting ruminal degradability and chemical composition of corn silage using near-infrared spectroscopy and multivariate regression

PONE-D-23-18553R2

Dear Dr. Pucetti,

We’re pleased to inform you that your manuscript has been judged scientifically suitable for publication and will be formally accepted for publication once it meets all outstanding technical requirements.

Kind regards,

Aziz ur Rahman Muhammad

Academic Editor

PLOS ONE

Additional Editor Comments (optional):

Dear Authors

Congratulation. Thanks for revising the manuscript according to suggestions
---

## [Editor Report · Acceptance letter]

28 Mar 2024

PONE-D-23-18553R2 

PLOS ONE

Dear Dr. Pucetti, 

I'm pleased to inform you that your manuscript has been deemed suitable for publication in PLOS ONE. Congratulations! Your manuscript is now being handed over to our production team.

Kind regards, 

on behalf of

Dr. Aziz ur Rahman Muhammad 

Academic Editor

PLOS ONE